# Performance of E-Banking and the Mediating Effect of Customer Satisfaction: A Structural Equation Model Approach

**Mohammed Arshad Khan** *[ID] and **Hamad A. Alhumoudi** [ID]

Department of Accountancy, College of Administrative and Financial Sciences, Saudi Electronic University, Riyadh 11673, Saudi Arabia; h.alhumoudi@seu.edu.sa
* Correspondence: m.akhan@seu.edu.sa

**Abstract:** Online payment is a trend that is gaining momentum globally. As a result of digitisation, the advent of online banking has increasingly made its way into the modern marketplace, serving not only customers but also corporations. The primary data were gathered from 287 participants. Stratified random sampling was used. Structure Equation Modelling (SEM), reliability, convergent, discriminate validity and model fitness were achieved through SmartPLS 3 (Christian M. Ringle, Germany). The findings reveal that efficiency, reliability and service quality have a significant direct effect on customer satisfaction and customer retention. It also shows the significant effect of efficiency, reliability and service quality when using customer satisfaction as a mediator for customer retention. It is possible that the data gathered may be valuable for both banks and enterprises interested in entering the Indian market. This research also specifies four main components of E-banking: efficiency, reliability, service quality and customer satisfaction.

**Keywords:** e-banking; reliability and efficiency; service quality; retention

## 1. Introduction

As one of the more recent e-banking services, online banking refers to any method that allows customers to conduct financial transactions electronically rather than in person at a physical location [1]. Technology's rapid evolution has had an impact on nearly every business, including banking, due to the obvious introduction of cutting-edge methods, including Internet banking, all around the world. The banking industry has seen enormous transformations, and India is no exception. Internet banking was made possible in part by India's economic globalisation in the 1990s. The Indian government underlined the financial industry's demand for digital banking to be responsible for moving out many committee recommendations on banking reforms [2].

That why (a) India's urban population consists of many employees who work in the information technology business, which means that they have an active Internet connection, and (b) the Indian population is more active in numerous professional initiatives around the world than is the case in other industrialised countries. There has been a huge amount of pressure in India to deploy e-banking [3]. To keep their customers safe from online fraudulent activity, the vast majority of Indian banks have now put into place user-friendly online banking facilities with high robustness. Most Indian Internet users choose to carry out their financial and banking transactions through the use of the Internet because of its convenience and time-saving features [4]. Nevertheless, online banking also puts clients at risk of scams they were not expecting. Two-step authentication is recommended by the Reserve Bank of India (RBI) to maintain the security level [5].

For both commercial and public sector banks, online banking is primarily meant to achieve two goals. The primary goal is to make customers' lives easier by meeting their needs, such as providing online access to account information, statement information, bill payment, money transfer [6] and the ability to register for new accounts and e-clearance

for things such as rent and loan payments [7]. The second goal is to lower operating expenses [8]. Online banking service quality, efficiency and dependability were studied in this research [9]. The literature on consumer satisfaction in online banking [10]. Particularly in India, is extremely sparse [11], despite the fact that customer satisfaction is multifunctional [12] and has been frequently explored in many circumstances [13].

It was the goal of this research to look into the aspects that may encourage people to utilise online banking [14]. There are many factors involved in the level of customer acceptability and satisfaction with online banking [15], including service quality; performance; reliability; protection; understanding; the quality of the Internet access; the time and money saved [16]; and the perceived advantages; convenience of use; belief; and customer attitude toward using computers [17]. There is a direct correlation between client satisfaction and a bank's perceived utility and trustworthiness [18]. While online banking has a number of advantages [19], such as a reduced risk of fraud and hacking [20], there are also some drawbacks [21].

## 2. Review of Literature

Throughout the last couple of years, banks have made a significant change from traditional banking to electronic banking [22]. E-banking services and the user base have grown as a result of trends in innovation and technology improvements [23], paving the way for banking services to become more advanced and accessible. Electronic banking services have recently made it possible for clients to monitor their financial accounts, credit payments, checking accounts, texts and emails, payment swaps and other organisations via their smart phones, relying on bank instructions [24]. Online banking and mobile banking are two terms commonly used to describe e-banking [25]. Banks can make their services more readily available to their consumers by using two different platforms: online banking and mobile banking [26]. Internet-connected computers are required for online banking, whereas wireless mobile devices are required for mobile banking [27]. While clients believe that customisation is the most significant aspect of the mobile banking business, this emphasises the contrast between the parameters for online and mobile banking services [28]. E- banking's critical feature is trustworthiness [29]. In spite of the widespread perception that Internet bank services are the least expensive, bank customers believe that digital banks are neglected [30].

E-banking is beneficial for both banks and their customers since it enables them to promote and provide services or products online in a more economical, speedier and simpler manner, because it allows users to conduct financial transactions online from any location worldwide [31]. The bank began communicating with clients through a high-quality online experience [32]; the digital banking platform has become a rivalry agent for banks due to their ability to obtain and keep customers [33] stressed that by exploiting the Internet, fresh and innovative ways businesses may increase their income through delivering value to customers can be achieved through the use of online portals, which can give companies and consumers more opportunities to participate [34].

In today's world, most buyers make purchasing selections based on online content that has been offered or exchanged by other customers [35]. Thus, online banking has a massive effect on the brand's recognition, feelings and thinking among consumers [36]; as we know that everyone generally accesses the Internet on a regular basis when they purchase a product. They assist in the spread of information through electronic means [2]. In [37], the association between the level of service and the likelihood of making a purchase was examined using data from 201 respondents; it was also revealed that pleasure acts as a facilitator between quality of service and purchasing intentions. [5] examined the same problem with a group of 708 individuals from diverse businesses and found that customer retention is positively impacted by the perception of service quality [38].

A focus on customer satisfaction and increased levels of product quality go hand in hand [22]. More critically, e-banking [39] has a greater impact on the development of the banking industry due to its reliability and tenability [40]. In all industries, but particularly

in the service industry, a company's ability to keep clients satisfied is critical. Customer satisfaction is contingent upon buyers paying for goods or services and utilising such goods or services [41]. The prosperity of a bank is directly related to the attitudes and preferences of its customers. Elevated electronic banking, corporate performance and customer objectives are all linked to customer satisfaction. An organisation's ability to grow and succeed depends on the satisfaction of its customers [42]. Quality of service is firmly connected to customer satisfaction. Due to the ease with which the advantages of similar services can be assessed, online banks tend to be more interested in client views of online banking services. The quality regime's absolute achievements are supported by customer satisfaction, the outcome of which is heavily influenced by the client's awareness of the total service quality [43].

## 3. Hypotheses

Efficiency and reliability are the capacity to consistently and correctly fulfil an agreed-upon task. Banks are expected to ensure dependability and consistency in fulfilling financial operations, according to [44], but it is also vital to depict this prominence using e-based services. Efficiency and reliability are also important aspects in assuring satisfaction and retaining customers. The customer specifies that their transaction be executed via e-banking, and effectiveness indicates that the maximum number of transactions are completed. [45] go on to say that dependability in an online activity can increase customer engagement with the service and motivate users to stay. In accordance with [46], when using any online service, a consumer must first verify the source's authenticity, impartiality and protection of individual details. Earlier studies have shown a favourable and statically relevant link amongst the efficiency and reliability and retention and satisfaction [47]:

**Hypothesis 1 (H1).** *Efficiency and reliability have a positive effect on customer satisfaction.*

**Hypothesis 2 (H2).** *Efficiency and reliability have a positive effect on customer retention.*

The quality of a company's service determines its sustainability in the marketplace [48]. Today's customers have different ideas about what constitutes good customer service than those who came before them. The quality of online banking services must therefore be evaluated. To determine the quality of online banking services, it is vital to identify the most relevant components, as well as how users evaluate computerised e-banking. Customer perspectives and the scope of the service of a bank are largely influenced by customer preferences. Customers' perceptions of the quality of their services, as well as their dedication and eagerness to assist the organisation with e-banking services, were examined by [49]. On the other hand, it is impacted by the customer's trust in online services and the security of their personal data. Many customers are hesitant about using e-banking services and prefer using branch services. According to [50], the fundamental indicator of customer satisfaction is service quality. If this criterion is unfulfilled, they face growing competition and greater customer access to better services while profit margins drop. Banks must emphasise customer satisfaction through increasing efficiency, reliability and customer service [51]:

**Hypothesis 3 (H3).** *Customer satisfaction is positively influenced by service quality.*

**Hypothesis 4 (H4).** *Service quality has a positive effect on customer retention.*

Customer satisfaction is critical for all organisations, but it is more critical for the service sector. Customer satisfaction is predicated upon customers purchasing goods or services and utilising them [52]. The perceptions of customers and their choices in terms of service quality have a massive effect on the bank's success. Advancements in electronic banking, company success and customer expectations are all related to customer satisfaction [53]. When consumers are satisfied, business performance improves, and the company

can expand. The quality of service and the level of customer satisfaction are inextricably linked [54]. Since the gains of competing services can be easily quantified, customer perceptions must be a higher priority for online banking institutions than their services. [55] say that customer satisfaction is the fundamental basis of the quality revolution's most remarkable milestones, which are essentially based on consumer knowledge of the entire service quality [56]:

**Hypothesis 5 (H5).** *Customer retention is positively correlated with customer satisfaction.*

**Hypothesis 6 (H6).** *Customer satisfaction mediates the association between efficiency and reliability and customer retention.*

**Hypothesis 7 (H7).** *Customer satisfaction mediates the association between service quality and customer retention.*

## 4. Research Methodology

The nature of this research study was descriptive-cum-cross-sectional. In order to assess the effects of efficiency and reliability on customer satisfaction and customer retention in India, we used a questionnaire and analysed the elements of e-banking services that would affect service quality, satisfaction and customer retention. The final questions are divided into two sections: basic knowledge about the customers and questions regarding the four criteria of e-banking services ("efficiency and reliability, service quality, customer satisfaction, and customer retention"). The Google Form was created to collect feedback from Indian participants. A "five-point Likert Scale" was used to assess client satisfaction, with 1 representing "Strongly Disagree" and 5 representing "Strongly Agree". The questionnaires were distributed to a total of 295 people in India, and a total of 287 were selected for analysis. Smart PLS 3 was applied to achieve convergent and discriminant validity, model fitness and Structure Equation Modelling (SEM).

## 5. Findings and Discussion

It consisted of 30 questions separated into two segments. One section addressed the respondent's demographic variables, while the other was divided into four e-banking service indicator effects of "efficiency and reliability, service quality, customer satisfaction, and customer retention". To assess the data from participants, a summarised approach of the "rating scale" was implemented. Although there were 295 respondents, exactly 287 were chosen for the research to facilitate for processing data. To perform a quantifiable examination of the data, the software "SmartPLS 3" was applied. This section contains the findings and results of the research.

### 5.1. Baseline Information of the Respondents

This part contains a sample of the persons who filled out the questionnaire. Table 1 summarises the responses on the basis of demographic characteristics that were chosen for the studies. The information offered in this section is derived from primary sources.

Table 1 depicts the demographic background of the participants, including their gender, age group, educational qualification, occupational position and monthly income, among other information. It reveals that males constituted the majority of sample respondents (60.27 percent), with females accounting for the remaining 39.73 percent of respondents. As indicated by the survey results, the vast majority of respondents (31.70 percent) were between the ages of 31 and 40, 30.31 percent were between the ages of 21 and 30, 27.87 percent were over 40, and 10.10 percent were between the ages of 21 and 20 years old.

**Table 1.** Baseline data of the participants (N = 287).

| Basis | Categories | F | % |
|---|---|---|---|
| Gender | M | 173 | 60.27 |
| | F | 114 | 39.73 |
| Age Group | Up to 20 years | 29 | 10.10 |
| | 21–30 years | 87 | 30.31 |
| | 31–40 years | 91 | 31.70 |
| | 41 and above | 80 | 27.87 |
| Educational Qualification | U.G | 41 | 14.28 |
| | G | 82 | 28.57 |
| | P.G | 91 | 31.70 |
| | P.D.H | 73 | 25.43 |
| Occupational Status | Govt. Employees | 72 | 25.08 |
| | Private Employees | 56 | 19.51 |
| | Business and self Employees | 112 | 39.02 |
| | Students | 47 | 16.37 |
| Monthly Income | ≤Rs 10,000 | 32 | 11.14 |
| | Rs 10000–Rs 20,000 | 87 | 30.31 |
| | Rs 20,001–Rs 40,000 | 102 | 35.54 |
| | >Rs 40,000 | 66 | 22.99 |

Following education, 14.28 percent of respondents were undergraduates (U.G), 28.57 percent had graduated (G), 31.70 percent were postgraduates (P.G), and 25.43 percent held a professional degree (PDH). Respondents were divided into four groups based on their employment status: government employees (25.08 percent), private employees (19.51 percent), business or self-employed (39.02 percent) and students (16.37 percent).

When it comes to monthly income, the results show that 11.14 percent of respondents had incomes under INR 10,000, 30.31 percent had incomes between INR 10,000 and 20,000, 35.54 percent had incomes between INR 2000 and 40,000, and 22.99 percent had incomes over INR 40,000.

*5.2. Measurement Model Evaluation*

The measuring model was confirmed using "internal consistency, convergent validity, and discriminant validity".

The circles in Figure 1 represent the latent constructs the researcher used in the study, which include "service quality, efficiency and reliability, customer satisfaction, and customer retention". There are five statement codes measuring service quality, three statement codes measuring efficiency and reliability, four statement codes measuring customer satisfaction and four statement codes measuring retention. They are depicted adjacent to the indicators leading to the corresponding items/constructs. For each item or construct, the factor loading values are displayed near the corresponding arrow.

Table 2 suggests that when the mean values of all items in a construct exceed 3, it implies a good response in "service quality, efficiency, and reliability to customer retention and satisfaction", as well as the mediator role of customer satisfaction. The researcher used a five-point "Likert scale," with values ranging from "Strongly Disagree" (1) to "Strongly Agree" (5). There are items in each construct that have factor loadings above the prescribed limit of 0.70. So, each statement clearly explains its own theoretically assumed construct.

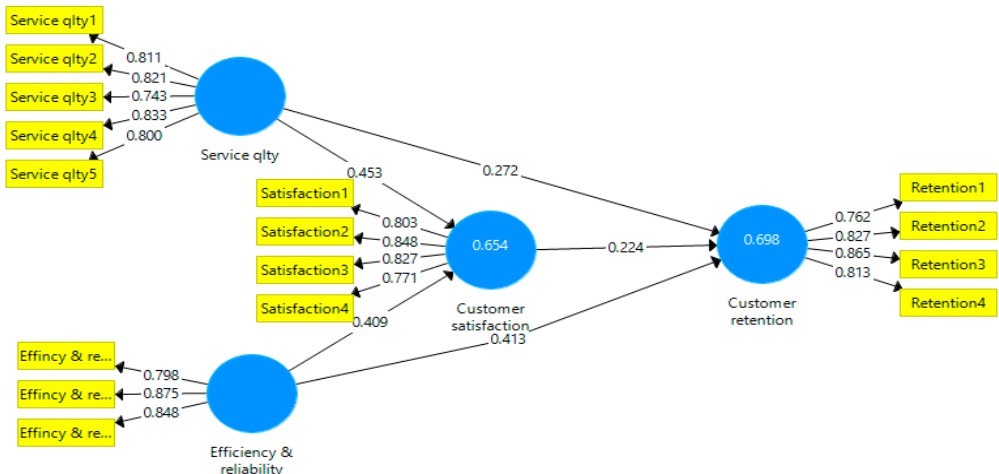

**Figure 1.** Measurement model from Smart PLS 3.0.

**Table 2.** Mean, SD and loadings of constructs.

| Construct | Item | Mean | SD | Loading |
|---|---|---|---|---|
| Efficiency and Reliability | Eff & reliab1 | 3.98 | 0.57 | 0.89 |
| | Eff & reliab2 | 3.78 | 0.65 | 0.76 |
| | Eff & reliab3 | 4.06 | 0.781 | 0.73 |
| Service Quality | Service quality 1 | 4.09 | 0.44 | 0.91 |
| | Service quality 2 | 3.11 | 0.63 | 0.92 |
| | Service quality 3 | 3.14 | 0.56 | 0.84 |
| | Service quality 4 | 4.05 | 0.88 | 0.73 |
| | Service quality 5 | 4.11 | 0.41 | 0.81 |
| Retention | Retention1 | 2.89 | 0.86 | 0.86 |
| | Retention2 | 3.01 | 0.71 | 0.79 |
| | Retention3 | 4.18 | 0.62 | 0.84 |
| | Retention4 | 4.21 | 0.63 | 0.90 |
| Satisfaction | Satisfaction1 | 2.98 | 0.71 | 0.81 |
| | Satisfaction2 | 3.03 | 0.55 | 0.91 |
| | Satisfaction3 | 4.24 | 0.62 | 0.72 |
| | Satisfaction4 | 4.01 | 0.91 | 0.76 |

As shown in Table 3, all five constructs met the required standards limit, as their "Composite Reliability" (C.R) values were greater than 0.7 and their "Average Variance Extracted" (A.V.E) values were greater than 0.5. Internal consistency was confirmed by "Cronbach's Alpha and rho-a" values that were significantly greater than 0.7. As a result, the concept of "convergent validity" was developed [57].

**Table 3.** Convergent validity result.

| Factor | Cronbach's Alpha | Rho-A | C.R | A.V.E |
|---|---|---|---|---|
| Efficiency and Reliability | 0.892 | 0.891 | 0.901 | 0.716 |
| Service Quality | 0.841 | 0.840 | 0.881 | 0.654 |
| Satisfaction | 0.818 | 0.817 | 0.863 | 0.681 |
| Retention | 0.864 | 0.865 | 0.889 | 0.698 |

## 5.3. Discriminant Validity Result

The discriminant validity was checked by applying the "Fornell-Larcker and cross-loading criteria". Discriminant validity specifies "the extent to which the measure is adequately distinguishable from related constructs within the nomological net".

To calculate the "Fornell-Larcker" criterion, we took the square roots of the "Average Variance Extracted" of the available constructs (see Table 4). To summarise, the values were as follows: efficiency and reliability were greater than the correlation coefficients amongst individual components (0.840), retention (0.818), satisfaction (0.813) and service quality (0.802). Finally, discriminant validity was established using the "Fornell-Larcker" criterion [58].

**Table 4.** Discriminant validity: Fornell–Larcker criterion.

| Factors | Customer Retention | Customer Satisfaction | Efficiency and Reliability | Service Quality |
|---|---|---|---|---|
| Customer Retention | **0.848** | | | |
| Customer Satisfaction | 0.733 | **0.823** | | |
| Efficiency and Reliability | 0.719 | 0.765 | **0.831** | |
| Service Quality | 0.768 | 0.774 | 0.762 | **0.809** |

Table 5 shows that the cross-loading criterion indicates that all constructions had higher loadings than cross-column cross-loadings of other structures. As a result, discriminant validity was determined using the cross-loading criterion [59].

**Table 5.** Discriminant validity: loading and cross-loading criteria.

| Factor | Efficiency and Reliability | Satisfaction | Retention | Service Quality |
|---|---|---|---|---|
| Efficiency& Reliability1 | **0.768** | 0.616 | 0.703 | 0.705 |
| Efficiency& Reliability2 | **0.865** | 0.707 | 0.681 | 0.563 |
| Efficiency& Reliability3 | **0.838** | 0.645 | 0.661 | 0.677 |
| Satisfaction1 | 0.616 | **0.813** | 0.692 | 0.716 |
| Satisfaction2 | 0.732 | **0.859** | 0.549 | 0.562 |
| Satisfaction3 | 0.560 | **0.837** | 0.688 | 0.593 |
| Satisfaction4 | 0.767 | **0.781** | 0.682 | 0.611 |
| Retention1 | 0.557 | 0.696 | **0.782** | 0.672 |
| Retention2 | 0.551 | 0.514 | **0.817** | 0.601 |
| Retention3 | 0.624 | 0.730 | **0.845** | 0.771 |
| Retention4 | 0.726 | 0.688 | **0.802** | 0.611 |
| Service quality1 | 0.608 | 0.633 | 0.580 | **0.802** |
| Service quality2 | 0.690 | 0.558 | 0.702 | **0.811** |
| Service quality3 | 0.663 | 0.643 | 0.694 | **0.762** |
| Service quality4 | 0.571 | 0.567 | 0.711 | **0.820** |
| Service quality5 | 0.730 | 0.651 | 0.635 | **0.769** |

## 5.4. Structural Equation Model

When analysing the structural model, it was necessary to take multicollinearity into consideration in order to obtain reliable results. It was found that the "Variance Inflation Factor" (VIF) ranged from 1.493 to 2.257, indicating that the model did not have any

multicollinearity. In order to determine the significance of the hypothesis to assess the model's structural integrity, bootstrapping was used (3000 resamples) to verify it.

It is clear from the below Figure 2 of PLS-SEM model that when the *t*-values exceeded the prescribed regression weight limit of 1.96, each path was significant at the significance level of 5 percent and above (i.e., the estimated path parameter was significant). The SEM model's results are shown in Table 6.

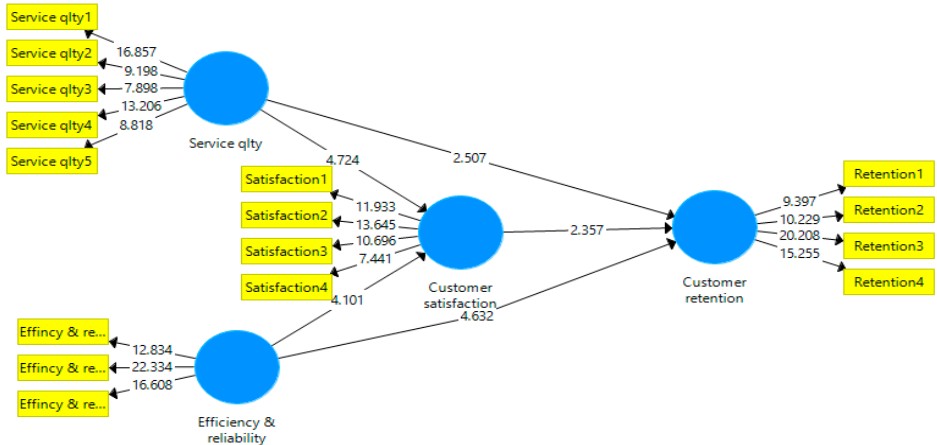

**Figure 2.** Structural Equation Model (SEM).

**Table 6.** Direct impact of service quality and efficiency and reliability.

| Hypothesis | Path | B | *t*-Value | *p*-Value | Result |
|---|---|---|---|---|---|
| H1 | Service Quality → Satisfaction | 0.443 | 4.624 | $p \leq 0.001$ | Supported |
| H2 | Service Quality → Retention | 0.265 | 2.607 | $p \leq 0.005$ | Supported |
| H3 | Efficiency and Reliability → Satisfaction | 0.410 | 4.301 | $p \leq 0.001$ | Supported |
| H4 | Efficiency and Reliability → Retention | 0.423 | 4.732 | $p \leq 0.001$ | Supported |
| H5 | Satisfaction → Retention | 0.234 | 2.457 | $p \leq 0.005$ | Supported |

Table 6 shows that hypotheses H1, H2, H3, H4 and H5 were supported. Service quality was also directly and positively related to consumer satisfaction (β = 0.443, *t*-value = 4.624 and *p* < 0.001). Service quality was also directly and positively related to consumer retention (β = 0.265, *t*-value = 2.607 and *p* < 0.005). Similarly, efficiency and reliability were directly and positively related to consumer satisfaction (β = 0.410, *t*-value = 4.301 and *p* < 0.001). Efficiency and reliability were also directly and positively related to consumer retention (β= 0.423, *t*-value = 4.732 and *p* < 0.001). However, satisfaction was directly and positively related to the retention of consumers (β = 0.234, *t*-value = 2.457 and *p* < 0.005).

Table 7 shows that hypotheses H06 and H07 were supported. In terms of service quality, satisfaction had a mediated or indirect effect on customer retention (β = 0.131; *t*-value = 2.312; *p* < 0.005). In the case of efficiency and reliability, satisfaction had a mediated or indirect effect on the customer's credibility (β = 0.095, *t*-value = 2.122 and *p* < 0.005).

**Table 7.** Mediating or indirect impact of satisfaction and hypothesis testing.

| Hypothesis | Path | B | *t*-Value | *p*-Value | Result |
|---|---|---|---|---|---|
| H6 | Service quality → Satisfaction → Retention | 0.131 | 2.312 | $p \leq 0.005$ | Supported |
| H7 | Efficiency and Reliability → Satisfaction → Retention | 0.095 | 2.122 | $p \leq 0.005$ | Supported |

## 6. Limitation of Study

Although the current study has some shortcomings, it can act as a basis for ongoing studies. First and foremost, the study's generalisability is limited, despite the fact that it is restricted to the banking industry. The study's results cannot be generalised to the rest of the industry, except for the banking sector. The second concern is that we only looked at one mediator variable—customer satisfaction—when additional factors may have been considered. The e-banking service could contain any additional appropriate moderating features to impact customer purchase intentions. E-banking service quality and customer retention could also be examined by researchers in future studies. A future study on client retention could follow a qualitative approach, or it could incorporate a range of methodologies and data kinds.

## 7. Implication of Study

The current paper contributes to the growing body of research in a number of ways. The model's properties can now be tested and validated in the Indian environment using the new TAM. The latest findings are crucial in that they provide insight into how customers view the degrees to which online banking is available to them. Furthermore, it provides a theoretical framework for examining online banking and technology issues. Educators can benefit from the study's findings as well. As a banker, you can use this research model to better understand how customers feel about online banking and how to improve Internet banking's features and function. The researchers say that by assessing potential modifiers, they have laid the groundwork for future improvements. Additionally, it is critical to build customer acceptance of e-banking's safety and security measures. Throughout India, Internet access is critical for online banking. Account security boosts confidence and client experience. According to the study, technology-enabled transactions bring convenience via multiple channels. The current study adds to the literature since it shows that perceived legitimacy, relative benefit and self-efficiency are contributing variables in adopting Internet banking.

## 8. Conclusions

The study looked at how customer satisfaction in Indian banking is affected by the quality of e-services. It is more probable that e-banking customers would stick around for the long haul if they have a positive experience. In this study, customer satisfaction was employed as a mediator to underline the link between satisfaction levels and future client retention. Efficiency and reliability, as well as customer satisfaction, were found to be critical mediators and predictors of client retention, particularly in the banking sector. Customers are sufficiently up to date with the latest developments in banking services and technology, and they have high expectations for the quality of the service they experience. The study's results will help us better grasp how banks work and take advantage of emerging information technology.

Banking service quality in India and other nations can be improved by establishing new services to meet customers' needs, and this approach can be effectively replicated in other nations. Customer satisfaction, retention, efficiency and reliability all have practical applications in banking. The Internet and the recent technical advances can help banks stay ahead of the competition, with a special emphasis on e-banking service quality and client touch points. The first goal is to develop and sustain a loyal customer base, and increasing the firm's capability to succeed is the next step to remain competitive in the market. Our study contributes to the existing body of expertise in the context of behavioural finance.

**Author Contributions:** M.A.K. and H.A.A. contributed to conceptualization, formal analysis, investigation, methodology, and writing and editing of the original draft. All authors have read and agreed to the published version of the manuscript.

**Funding:** This research received no external funding.

**Institutional Review Board Statement:** Not applicable.

**Informed Consent Statement:** Informed consent was obtained from the respondents of the survey.

**Data Availability Statement:** The data used to support the findings of this study are available from the corresponding author upon request.

**Conflicts of Interest:** The authors declare no conflict of interest.

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
