# Peer review of "Performance of E-Banking and the Mediating Effect of Customer Satisfaction: A Structural Equation Model Approach"

_sustainability, doi:10.3390/su14127224_

Round 1

Reviewer 1 Report

Reviewer

English language and style

( ) Extensive editing of English language and style required
( ) Moderate English changes required
(x) English language and style are fine/minor spell check required
( ) I don't feel qualified to judge about the English language and style

Yes

Can be improved

Must be improved

Not applicable

Is the content succinctly described and contextualized with respect to previous and present theoretical background and empirical research (if applicable) on the topic?

( )

(x)

( )

( )

Are the research design, questions, hypotheses and methods clearly stated?

( )

( )

( )

( )

Are the arguments and discussion of findings coherent, balanced and compelling?

( )

( )

( )

( )

For empirical research, are the results clearly presented?

( )

( )

( )

( )

Is the article adequately referenced?

( )

(x)

( )

( )

Are the conclusions thoroughly supported by the results presented in the article or referenced in secondary literature?

( )

(x)

( )

( )

Comments and Suggestions for Authors with minor revision.

The paper identifies the Performance of E-Banking and the mediating effect of customer satisfaction: a structural equation model approach. The subject is very interesting and relevant. It is an original and well-structured paper. The writing performance of the text is almost good with a few spelling, grammar and punctuation mistakes. Further, the text is attractive (engaging), friendly, easy to a reader to understand, and it makes the right impression on a reader. However, there are few readability problems as does not provide all the elements such as: For the survey, I found a total of 295 responses, but only 287 were chosen for the study to facilitate data processing. It is a survey paper based on the assessments of the participants. SmartPLS tool is used to perform the experiments.  In this regard, i could not find the basic statistical analysis experiments like descriptive analysis, T-Test, Z-Test, covariance analysis, and regression analysis. The authors also did not define input, mediator, and output variables through the model.

Author Response

 Response to Reviewer Comments and Suggestions

First of all, we pay our gratitude to the reviewer for his/her valuable comments. As kindly suggested by the esteemed reviewer, the present study has been thoroughly revised. This is for the favour of publication in your esteemed journal. The point-by-point response is given below.

Comment 1: For the survey, I found a total of 295 responses, but only 287 were chosen for the study to facilitate data processing.

Response 1: We are highly obliged to the reviewer for highlighting this aspect. I found a total of 295 responses, but only 287 were chosen for the study to facilitate data processing because 7 respondents (295-287) did not respond properly. So, we do not consider those 7 responses in this paper and eliminate them.

Comment 2: It is a survey paper based on the assessments of the participants. SmartPLS tool is used to perform the experiments.

Response 2: We are very thankful to the reviewer for identifying this important aspect. It is a perception that the SmartPLS tool is used to perform the experiments. But it is not true that SmartPlS developers mostly used this software in survey-based research or marketing-based research. So, it can be applicable for experiments as well as survey based.

Comments 3: In this regard, i could not find the basic statistical analysis experiments like descriptive analysis, T-Test, Z-Test, covariance analysis, and regression analysis.

Response 3: We are highly obliged to the reviewer for highlighting this aspect. We are highly obliged to the reviewer for highlighting this aspect. We used descriptive analysis, mean and standard deviation. We also used t-test and structural equation modelling (SEM) for that type of regression analysis. We will definitely consider these remaining tests in future research.

Comment 4: The authors also did not define input, mediator, and output variables through the model.

Response 4: We are very thankful to the reviewer for identifying this important aspect. Input, mediator, and output variables are defined in figure-1 of Chapter 6.

Reviewer 2 Report

The aim of the papier was looking into the aspects that may encourage people to utilise online banking. Author has provided a short literature overview.

The nature of this research study was descriptive-cum-cross sectional. In order to assess the effects of efficiency and reliability on customer satisfaction, and customer retention in India, a questionnaire and analysed the elements of e-banking services that would affect service quality, satisfaction, and customer retention were used.

The primary data was gathered from 287 participants where use of stratified random sampling was imposed.

On the base of literature description he pointed out 7 hypothesis. They were verified with Structure Equation Modelling (SEM), Reliability, Convergent, Discriminate Validity and model fitness are achieved through SmartPLS 3.

At the same time, the author emphasized limitations and implication of undertaken research.

At the same time, the author emphasized limitations and implication of undertaken research. First and foremost, the study's results cannot be generalised to the rest of the industry, except from the banking sector. The second concern is that we only looked at one mediator variable - customer satisfaction- when additional factors may have been considered.

The conclusions are consistent with the evidence and arguments presented. 

I recommend the article for publication.

General outlines:

The literature review is relevant to the topic. Most of the cited references are from the last five years.  The citations of older publications give a good background for the discussed issues.

The tables and figures presented in the article enrich the discussed issues and support the description of the presented issues.

Few mistakes:

- missing punctuation marks.

- names of the citied authors  in capital litters. P. 3 (MUHAMMAD & RANA) p. 3 and 4 (MINHAJ);

- please check the list of the references – missing papers, for example : Muhammad and Rana or differences in item descriptions for example: M. A. Khan and S. M. MINHAJ, “Performance of online banking and direct effect of service quality on consumer retention and credibility of consumer and mediation effect of consumer satisfaction.,” Int. J. Bus. Inf. Syst., vol. 1, no. 1, p. 1, 2021, doi: 10.1504/IJBIS.2021.10043829.

Author Response

Response to Reviewer Comments and Suggestions

First of all, we pay our gratitude to the reviewer for his/her valuable comments. As kindly suggested by the esteemed reviewer, the present study has been thoroughly revised. This is for the favour of publication in your esteemed journal. The point-by-point response is given below.

Comments 1: The literature review is relevant to the topic. Most of the cited references are from the last five years.  The citations of older publications give a good background for the discussed issues.

Response 1: Thanks for the suggestion. We add some old citations and references, which we highlight in yellow.

Comment 2: names of the citied authors in capital litters. P. 3 (MUHAMMAD & RANA) p. 3 and 4 (MINHAJ);

Response 2: Thanks for the suggestion. We used Mendeley software for citation. It is not possible to change these citations manually.

Comment 3: please check the list of the references – missing papers, for example : Muhammad and Rana or differences in item descriptions for example: M. A. Khan and S. M. MINHAJ, “Performance of online banking and direct effect of service quality on consumer retention and credibility of consumer and mediation effect of consumer satisfaction.,” Int. J. Bus. Inf. Syst., vol. 1, no. 1, p. 1, 2021, doi: 10.1504/IJBIS.2021.10043829.

Response 3:  Thanks for the suggestion. We provide citations and references.

(MUHAMMAD & RANA, 2020).

MUHAMMAD, A., & RANA, A. H. (2020). Impact of Online Customer Relationship Management (OCRM) Upon Customers Satisfaction in Post Covid-19 Scenario. A Case Analysis of Standard Chartered Bank Pakistan. International Review of Management and Business Research, 9(4), 180–195. https://doi.org/10.30543/9-4(2020)-16

(Mohd Altaf Khan & MINHAJ, 2021)

  1. A. Khan and S. M. MINHAJ, “Performance of online banking and direct effect of service quality on consumer retention and credibility of consumer and mediation effect of consumer satisfaction.,” Int. J. Bus. Inf. Syst., vol. 1, no. 1, p. 1, 2021, doi: 10.1504/IJBIS.2021.10043829. https://www.inderscience.com/info/ingeneral/forthcoming.php?jcode=ijbis